# Nivel Corona Cohort: A description of the cohort and methodology used for combining general practice electronic records with patient reported outcomes to study impact of a COVID-19 infection

**Renee Veldkamp**[1]*, **Karin Hek**[1], **Rinske van den Hoek**[1], **Laura Schackmann**[1,2], **Eugène van Puijenbroek**[2,3], **Liset van Dijk**[1,2]

**1** Nivel, Netherlands Institute for Health Services Research, Utrecht, The Netherlands, **2** Department of Pharmacotherapy, -Epidemiology and -Economics, Groningen Research Institute of Pharmacy, University of Groningen, Groningen, The Netherlands, **3** Netherlands Pharmacovigilance Centre Lareb, 's-Hertogenbosch, The Netherlands

* r.veldkamp@nivel.nl

**Data Availability Statement:** The data is available at DANS (data archiving and networked services) at the DOI: 10.17026/dans-xzv-32r2.

## Abstract

### Aim

A population-based COVID-19 cohort was set up in the Netherlands to gain comprehensive insight in the short- and long-term effects of COVID-19 in the general population. The present study aims to describe the methodology and infrastructure used to recruit individuals with COVID-19, and the representativeness of the population-based cohort. The second aim was to characterize the population by description of their symptoms and health care usage during the acute COVID-19 phase.

### Method

The starting point of the set-up of the cohort was to recruit participants in routinely recorded, general practice electronic health records (EHR) data, which are sent to the Netherlands Institute for Health Services Research Primary Care Database (Nivel-PCD) on a weekly basis. Patients registered with COVID-19 were flagged in the Nivel-PCD based on their COVID-19 diagnoses. Flagged patients were invited for participation by their general practitioner via a trusted third party. Participating patients received a series of four questionnaires over the duration of one year allowing for a combination of data from patient reported outcomes and EHRs.

### Results

In this study, results from the first questionnaire are used. The Nivel Corona Cohort consists of 442 participants and is population-based, containing a complete image of severity of symptoms from patients with none or hardly any symptoms to those who were hospitalized due to the COVID-19. The five most prevalent symptoms during the acute COVID-19 phase

**Funding:** The setup and start of the study was not externally funded. Funding from ZonMw (project number 10430302110004) allowed for an extension of the study including two additional questionnaires and additional recruitment of participants. The funders had no role in study design, data collection and analyses, decision to publish, or preparation of the manuscript.

**Competing interests:** The authors have declared that no competing interests exist.

were fatigue (90.5%), reduced condition (88.2%), coughing/sneezing/stuffy nose (79.3%), headache (75.4%), and myalgia (66.7%).

## Conclusion

The population-based Nivel Corona Cohort provides ample opportunities for future studies to gain comprehensive insight in the short- and long-term effects of COVID-19 by combining patients' perspectives and clinical parameters via the EHRs within a long-term follow-up of the cohort.

## Introduction

The novel coronavirus, severe acute respiratory syndrome coronavirus 2 (SARS-CoV-2), detected in December 2019 has caused a pandemic with huge societal impact worldwide. Currently, the number of persons with a confirmed diagnosis of coronavirus disease 2019 (COVID-19) increased to 237 million individuals globally, with 4.8 million deaths (October 2021, [1]). Initial governmental, public health, and scientific responses focused on reducing the spread of COVID-19 and the burden on health care systems, treatment of patients with most life-threatening symptoms, and development of vaccines.

As the pandemic continued, a growing body of evidence indicated that COVID-19 both on the short and on the longer run differentially affected people. While some people were admitted to the intensive care unit, others experienced hardly any complaints during the acute phase. Moreover, a portion of those who recovered from COVID-19 developed persistent or even new symptoms lasting for weeks or months [2, 3]. Notably, persistent or new symptoms have also been reported by persons who had mild symptoms during the acute infection [4–6].

Furthermore, in order to gain in depth understanding of the course, severity, and short- and long-term impact of COVID-19 in the general population both patients' perspectives and clinical parameters are needed. These types of data provide valuable information such as: effects of a SARS-CoV-2 infection on activities of daily living, quality of life, lifestyle, and work on the one hand, and information on health care usage, treatment, and morbidity on the other hand. In the Dutch health care system, the general practitioner has a gate keeper function and general practices have defined lists of patients they care for, together covering all inhabitants of the Netherlands. As such, this system provides the unique possibility to reach and invite patients from the general population. Additionally, it allows for a linkage between patients' clinical information from electronic health records (EHR) of general practices, comprising complete records on morbidity and medication at patient level, with patient-reported outcomes using questionnaires [7]. The EHR also enables to make comparisons with the overall population and to check representativeness of responders.

With the aim to gain comprehensive insight in the short- and long-term effects of COVID-19 in the general population, a population-based COVID-19 cohort was set up in the Netherlands. A combination of patients' perspectives and clinical parameters within a long-term follow-up of the cohort allows for a thorough study on severity and duration of symptoms, pathophysiology, risk factors, and health care pathways of patients with post-acute COVID-19 syndrome. The aim of this study was twofold. The first aim was to describe the methodology and infrastructure used in the study to recruit individuals with COVID-19 and the representativeness of the population-based cohort. The second aim was to characterize the population-based cohort of COVID-19 patients, by a description of their symptoms and health care usage

during the acute phase as a starting point for further publications on the short- and long-term impact of a COVID-19 infection.

## Method

### Medical ethical committee

The Medical Ethical Committee (METC) of the VU Academic Medical Centre (VUMC) concluded that this study was not a clinical research with human subjects as meant in the Medical Research Involving Human Subjects Act (WMO) and the study was approved by the METC of the VUMC (METC protocol number 2020.0709) as well as by the applicable bodies of Nivel-PCD according to the governance code of Nivel-PCD (NZR-00320.081). All participants received written information on the study and gave digital informed consent. Participants could additionally provide informed consent for linkage of EHR data to questionnaire data. No consent from parents or guardians was needed for the minors in this study as they were all 16 years and older.

### Setting

Nivel, The Netherlands Institute for Health Services Research, has a Primary Care Database (Nivel-PCD) in which electronic health records (EHR) of general practices are collected. Approximately 500 general practices, covering around ten percent of the Dutch population, participate in the Nivel-PCD. Data in Nivel-PCD are pseudonymized at the source (in the practices), leaving out directly identifying data such as name or address. Nivel-PCD receives data on a yearly basis, and from a subset of approximately 350 practices also weekly. In 2016 the Benefit, Risk and Impact of Medication Monitor (BRIMM) research infrastructure was developed [7]. This infrastructure was used in the present study for recruitment (explained below) and allowed research combining data from EHRs and patient reported outcomes (PROs).

### Cohort recruitment

Fig 1 shows an overview of the recruitment process. In an early enquiry in May 2020 among practices participating in Nivel-PCD to assess feasibility of the study, 90 practices had already shown their interest in the study. A selection of 25 general practitioners (GPs) was invited via e-mail to participate in this study in February 2021. Practices were selected based on completeness of morbidity data in 2019, having weekly data in 2020, using R83.03 to code COVID-19 cases and having sufficient COVID-19 cases. The data received by Nivel-PCD consists of routinely registered health care data by the GP, among which diagnoses. International Classification of Primary Care (ICPC) codes are used to code diagnoses. The ICPC-code R83.03 was introduced by the Dutch College of General Practitioners to register COVID-19 from November 2020 onwards. The diagnosis of COVID-19 for an individual patient could be in their EHR when the patient consulted their GP directly, or when the patient contacted the Municipal Health services (GGD), who provided the national testing facilities. The GGD sent information on positive tests to GPs via automated coupling under the prerequisite that patients gave consent. An algorithm was created to scan the EHR of the participating GPs to flag pseudonymized persons of 16 years and older that have had COVID-19 based on the ICPC-code R83.03.

The flagged pseudonymized patient-ids were sent to a trusted third party (TTP), which held the encrypted key between the pseudonymization and the accompanying patient. The TTP sent a list of the flagged patients to the associated GP. The GP subsequently checked whether the flagged patients were eligible for the study. Patients were excluded when they were not proficient in the Dutch language, had passed away, were terminally ill, had moved or when they

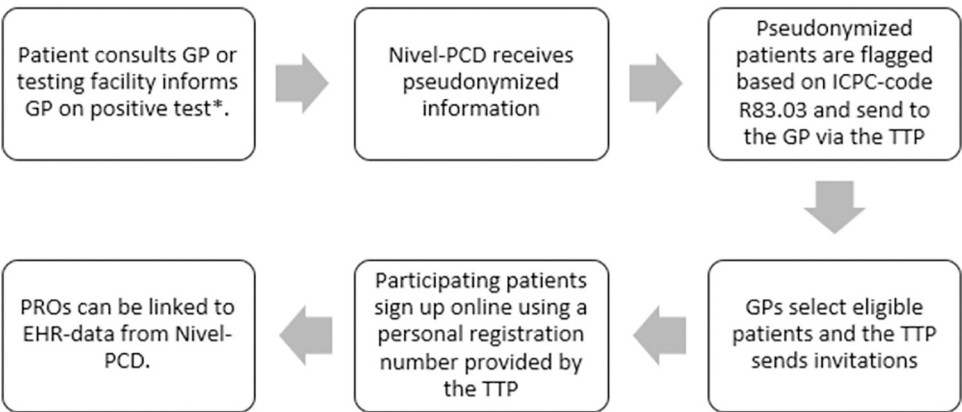

**Fig 1. Cohort recruitment via BRIMM infrastructure.** Abbreviations: GP: general practitioner, Nivel-PCD: Nivel Primary Care Database, ICPC: international classification of primary care, TTP: trusted third party, PROs: patient reported outcomes, EHR: electronic health records. Adapted from Hek et al. (2022) [7]. * The Municipal Health services (GGD) provided the national testing facilities and positive tests were sent to GPs via automated coupling under the prerequisite that patients gave consent.

were judged by their GP as not having had COVID-19 or not being capable of filling out questionnaires, for example due to cognitive deficits, illiteracy, personal problems, or too severe disease burden. The GPs were asked to register the reason for exclusion.

Thereafter, the GP provided names and addresses of all patients deemed eligible for participation to the TTP. The TTP sent an invitation letter on paper on behalf of the GP, containing information on the study and the question to participate. A reminder letter on paper was sent three weeks after the invitation. Patients could participate using a personal registration number provided by the TTP which was not available to the researchers, who only received pseudonymized data.

**Timeframe.** See Fig 2A for a visualization of the timeline. Scanning of the EHR of the participating GPs and inviting potential participants started in April 2021. In April 2021 patients

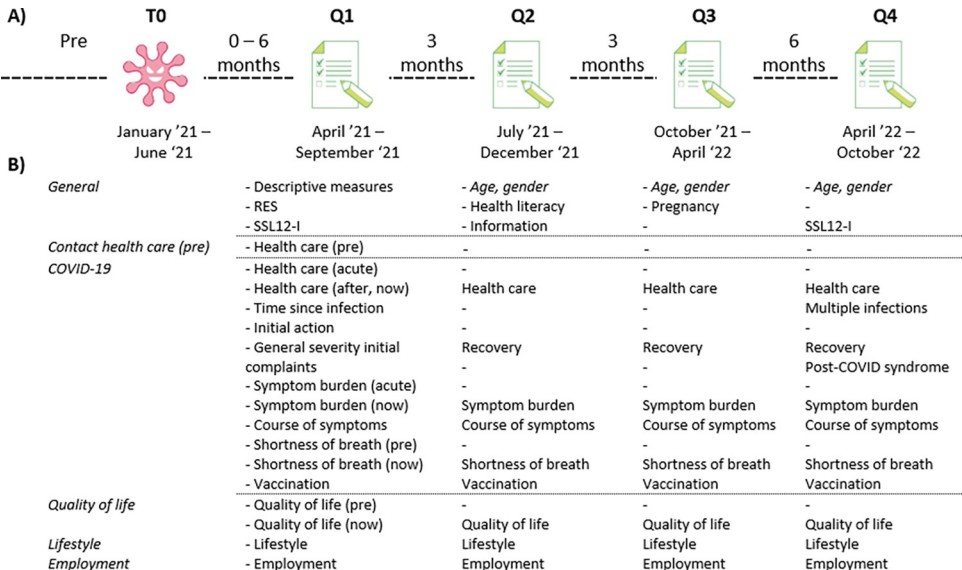

**Fig 2. Overview of the timeline of questionnaires in the Nivel Corona Cohort.** Abbreviations: RES: resilience evaluation scale, SSL12-I: Social Support List Interaction version.

with a COVID-19 registration, in a timespan of six weeks prior, were flagged and sent to the TTP. This process was thereafter updated and repeated every two weeks until July 2021. A last recruitment round among the participating practices was performed in July 2021, flagging all patients with a COVID-19 registration in the first three months of 2021. Therefore, the time between the infection and the response on the first questionnaire (Q1) could differ from zero to six months between respondents (Fig 2A).

The participants filled out four questionnaires, the first directly after agreement to participate (Q1), then subsequently after three months (Q2), after six months (Q3), and after one year (Q4). Therefore, Q1 does not refer to the onset of the COVID-19 infection for the patient, but to the beginning of participation of the patient (i.e., between zero and six months after infection). Participants noted their age and gender in each questionnaire as a control measure.

## Patient reported outcomes

The first questionnaire was the most extensive (see Fig 2B for an overview and S1A Table for an extensive description of the questionnaires). Questions were ordered by topic and consisted of information on determinants and outcomes for in depth studies on short- and long-term impact of COVID-19. The following determinants were assessed: descriptive data, including marital status and migration background, life style measures, such as BMI (height and weight) and smoking status, severity of the COVID-19 infection and resilience and received social support. The following outcomes were measured: perceived symptoms and their severity, information on (the impact of COVID-19 on) health- and self-care, quality of life, and employment. For a study on trust in information we included questions on information provision during the COVID-19 pandemic. For the current study we used data on the acute COVID-19 infection to describe the cohort.

## Information drawn from the electronic health records

A description of data present in the electronic health records is given in S1B Table. For the current study we used information on age, sex, and number of chronic diseases as defined by Nielen et al (2016) [8] from the EHRs. Additionally, the presence of comorbidities that were known risk factors for more severe COVID was based on registered ICPC-codes in the EHRs, namely diabetes mellitus (ICPC T90), fat metabolism disorder (ICPC T93), lung disease (ICPCs R28, R91, R95, R96), and hypertension (ICPC K86, K87).

## Data analysis

All analyses were conducted in Stata v16 (StataCorp. 2019. *Stata Statistical Software*: *Release 16*. College Station, TX: StataCorp LLC). Missing values were not imputed and significance level was set at 0.05. First, representativeness of the cohort was assessed. The participants of the Nivel Corona Cohort were compared to the patients that were invited but who did not participate and that were flagged (selected on age and ICPC-code R83.03) but who did not participate. The groups were compared on the basis of age, sex distribution, number of chronic diseases and the presence of diabetes, fat metabolism disorder, lung disease and hypertension, by using Wilcoxon rank-sum tests for the non-normally distributed continuous variables and Chi-square tests for the categorical variables. Then we provided a characterization of the cohort by describing the symptoms and health care usage in the acute phase of the infection. To this end, data from Q1 were used. Descriptive data on general characteristics of the cohort and outcomes of the PRO (Q1) were given by providing means and standard deviations or frequencies and percentages as appropriate. As not all participants gave an answer on all questions, the total number of respondents can differ per question.

# Results

## GP and patient participation

Of the 25 invited general practices, 18 GPs (72%) participated in this study, comprising 91,776 listed patients (median of 3,804 per practice). Practices were spread throughout the Netherlands (one practice in the North, five in the East, eight in the West, and four in the South) and located in both rural and urban areas.

In total 2,103 individuals aged 16 years and older with COVID-19 registration were flagged in the system and sent to the GPs. The GPs excluded 252 of these individuals (12.0%) and provided reasons for 58.7% of them. The most frequent given reasons were not sufficiently understanding the Dutch language (17.9%), being passed away (9.1%), admitted to a nursing home (6.7%), or it was unknown whether the patient had COVID-19, the patient had no COVID-19, or the infection was too long ago (8.4%). Ultimately, 1,851 individuals (88.0%) were invited to participate in the study of whom 448 individuals agreed to participate, of which one indicated to not have had COVID-19 and five withdrew their participation almost immediately. These six persons were therefore excluded from further analyses, leaving 442 respondents in the final Nivel Corona Cohort of whom 421 participants (95.2%) consented for linkage of questionnaire data to their EHR.

## Representativeness of the cohort

Table 1 shows comparisons between the Nivel Corona Cohort (n = 442) and the group of invited patients that did not participate (n = 1851–442 = 1409) and the group of flagged patients that did not participate (n = 2103–442 = 1661). Patients in the Nivel Corona Cohort were significantly older, more often female and less frequently had lung diseases than the invited and flagged non-participating patients. No differences between the cohorts were found on number of chronic diseases and presence of diabetes, fat metabolism disorder, or hypertension.

## Description of the cohort

**General characteristics.** Table 2 shows the descriptive measures for the final Nivel Corona Cohort. The average age of the sample was 51.5±13.4 years (n = 439, range 17–89 years), with the majority being female (60.9%) and middle (188, 47.0%) or highly (164, 41.0%) educated. Most of the respondents were married (295, 72.0%), and the majority cohabited

**Table 1. Representativeness of the Nivel Corona Cohort.**

|  | Nivel Corona Cohort | Invited patients[a] | *p*-value | Flagged patients[b] | *p-value* |
|---|---|---|---|---|---|
| N | 442 | 1409 |  | 1661 |  |
| Age | 51.5 ± 13.4 | 45.7 ± 17.3 | < .001* | 46.5 ± 18.1 | < .001* |
| Sex (% F) | 60.9% | 52.2% | 0.001* | 52.6% | 0.002* |
| Nr. of chronic diseases | 1.3 ± 1.7 | 1.3 ± 1.8 | 0.973 | 1.4 ± 1.9 | 0.356 |
| Diabetes (% yes) | 4.6% | 5.8% | 0.340 | 6.8% | 0.102 |
| Fat metabolism disorder | 7.3% | 6.0% | 0.370 | 6.4% | 0.542 |
| Lung disease | 10.9% | 15.4% | 0.022* | 15.5% | 0.018* |
| Hypertension | 14.0% | 11.8% | 0.222 | 13.0% | 0.569 |

[a] Invited patients that did not participate in the Nivel corona cohort.

[b] Patients aged 16 years and older that were selected from the electronic health record with ICPC-code R83.03 and did not participate in the Nivel corona cohort.

*p*-value < 0.05. *Abbreviations*: F: female; N: number.

**Table 2. Descriptive characteristics of the Nivel Corona Cohort at Q1.**

| | |
|---|---|
| Age (n = 442) | 51.5 ± 13.4 |
| Sex (n = 442) | |
| Female | 269, 60.9% |
| Male | 172, 38.9% |
| Other | 1, 0.2% |
| BMI (kg/m$^2$) | 26.1 ± 4.3 |
| Education (n = 400) | |
| Low | 48, 12.0% |
| Middle | 188, 47.0% |
| High | 164, 41.0% |
| Marital status (n = 410) | |
| Married | 295, 72.0% |
| Divorced | 38, 9.3% |
| Widow(er) | 9, 2.2% |
| Never married | 68, 16.6% |
| Living with (n = 410) | |
| Family* | 197, 48.1% |
| Partner | 127, 31.0% |
| Children | 20, 4.9% |
| Parents | 20, 4.9% |
| Alone | 42, 10.2% |
| Others | 4, 1.0% |
| Country of birth NL (n = 411) | |
| Respondent | 394, 95.9% |
| Mother | 388, 94.6% |
| Father | 384, 93.2% |

*With partner and children. *Abbreviations*: BMI: Body Mass Index; NL: Netherlands; n: number.

with their partner, (and) children or with their parents/caretakers (364, 88.8%). Lastly, most respondents were born in the Netherlands, as well as their mother and father (95.9%, 94.6% and 93.2%, respectively).

## The COVID-19 infection

At the time of the Q1 the majority of respondents indicated the infection to be one to three months ago (n = 261, 59.1%). Others responded it was shorter than a month ago (n = 15, 3.4%), three to six months ago (n = 52, 11.8%) or longer than six months ago (n = 114, 25.8%). Only 6.3% (n = 28) of the respondents indicated that they got COVID-19 after receiving a vaccination.

During the acute COVID-19 phase, 90 respondents (20.4%) indicated to have had none or hardly any symptoms, 158 (35.8%) indicated to have had symptoms comparable to a severe cold, 160 (36.3%) had many symptoms but did not need to go to the hospital, and 33 (7.5%) had so many symptoms that they were hospitalized. In accordance, of the 441 respondents, 29 (6.6%) respondents needed to go to the nursing ward of the hospital, nine (2.0%) to the intensive care, and seven (1.6%) needed to go to a rehabilitation center for a longer duration than one day. Further, 64 respondents (14.5%) went to the GP, 20 (4.5%) to the out-of-hours primary care services, 21 (4.8%) to the first aid, five (1.1%) had a day treatment in the rehabilitation center, and two (0.5%) received home care.

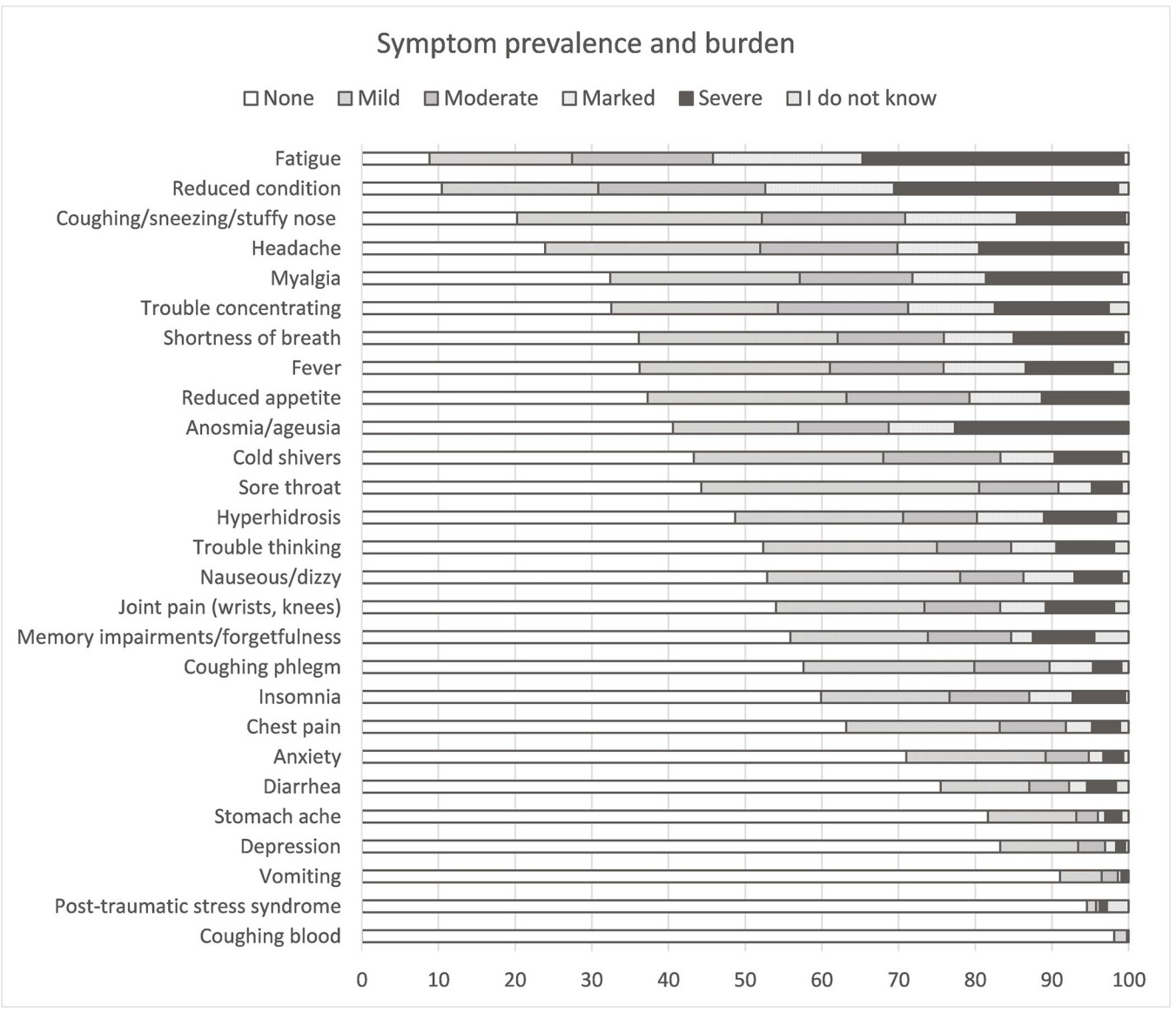

**Fig 3. Prevalence and burden of symptoms during the acute COVID-19 infection.**

Of 423 respondents, 50 (11.8%) received antibiotics from their GP and 84 (19.9%) received other medicine(s) for their COVID-19, such as dexametason (n = 11), paracetamol (n = 10) and prednisone (n = 9).

Fig 3 provides an overview of the prevalence and severity of symptoms during the acute COVID-19 phase. The ten most prevalent self-reported symptoms were fatigue (90.5%, mild, moderate, marked or severe), reduced condition (88.2%), coughing/sneezing/stuffy nose (79.3%), headache (75.4%), myalgia (66.7%), trouble concentrating (64.9%), shortness of breath (63.2%), reduced appetite (62.5%), fever (61.7%), and anosmia/ageusia (59.2%). The symptoms for which most respondents indicated that the symptom burden was marked to severe were fatigue (53.5%), reduced condition (46.0%), anosmia/ageusia (31.1%), headache (29.5%), and coughing/sneezing/stuffy nose (28.7%) (see also S1 Fig).

The largest proportion of respondents (see S2 Table) indicated to have had the following five symptoms with a duration of more than a week during the acute infection: fatigue (n = 344, 77.8%), reduced condition (n = 337, 76.2%), shortness of breath (n = 227, 51.4%), coughing, sneezing and/or stuffy nose (n = 221, 50.0%), and trouble concentrating (n = 207, 46.8%). When looking within the subgroup of those who had experienced a specific symptom, the most common symptoms that lasted longer than a week were reduced condition (n = 337, 86.6%), fatigue (n = 344, 86.2%), shortness of breath (n = 227, 81.7%), memory impairments/forgetfulness (n = 148, 88.1%), and post-traumatic stress syndrome (n = 9, 81.8%).

## Discussion

With the aim to gain comprehensive insight in the short- and long-term effects of COVID-19 in the general population, a population-based COVID-19 cohort was set up. Here, we provided a description of the representativeness of the cohort, and a characterization of the patients and the acute phase of their COVID-19 infection. Participants were recruited via EHRs of GPs and all Dutch inhabitants are registered in one general practice. This resulted in a population-based cohort consisting of a heterogeneous group of persons that had COVID-19. The Nivel Corona Cohort thereby represents individuals with COVID-19 in the general population by containing a complete range of number and severity of symptoms. The cohort provides the opportunity to study the short- and long-term effects of an infection with COVID-19 in a population-based sample instead of mainly in patients who were hospitalized [9]. It further provides the opportunity to use a combination of patients' perspectives and clinical parameters from EHRs.

However, it should be noted that participants in the present cohort were in general a bit older, slightly more often female, and less frequently had lung disease comorbidities than the patients invited or flagged for the study, but who did not participate. Notably, it is important to take into account that the participants in the present study had COVID-19 in the first half of 2021. In this period the Netherlands experienced the third wave and the alpha and later delta variant were the most prevalent [10]. At that moment in time, two million Dutch inhabitants had been diagnosed and registered with COVID-19 [11] and approximately 32,000 people had died in the Netherlands with COVID-19 as registered or probable cause of death [12]. Additionally, this was at the start of the nationwide vaccination rollout and the majority of participants got COVID-19 before they received the vaccine.

The general image of the acute infection was in accordance with previous studies [9, 13], with the most common symptoms being fatigue, reduced condition, coughing, sneezing and/or stuffy nose, headache and myalgia. The heterogeneity in severity of symptoms and in health care usage during the acute infection depicts recruitment of a population-based cohort.

### Strengths and limitations

The present study gives a description of the Nivel Corona Cohort and the initial effects of the infection. The set-up provides the opportunity for a thorough longitudinal study in a population-based cohort wherein perceptions of patients can be combined with data from EHRs. Therefore, future papers using this cohort will focus on the course of the symptoms and their effects on quality of life, employment, and clinical information from EHR on prescriptions, GP consultations, referrals and lab measurements. Furthermore, using EHR data allows for example for analysis of the impact of prior medical history on the severity of COVID-19 and on the relation with quality of life. In addition, care pathways can be traced and longitudinal patterns of symptoms presented in general practice can be described.

However, some limitations of the current cohort should be taken into account. First, there is no control group to compare the answers of the respondents and the PROs. It is therefore not precluded that results on the PROs are (partly) an effect of the pandemic and its lockdowns, rather than of the respondents getting COVID-19 themselves. Secondly, for some questions respondents were also asked to describe the situation before the infection, as for example the SF12. Therefore, some recall bias might be present. Furthermore, there was a broad range in the time between the infection and the first questionnaire, which needs to be accounted for when analyzing the post infection questions. Lastly, as potential participants were judged for eligibility by their GP, the recruitment depended on the personal views of the GP. Additionally, we did not offer the alternative of paper questionnaires. Therefore, some groups were not represented by the cohort (e.g. those illiterate or digital illiterate, not sufficient proficient in the Dutch language, too severe disease burden). This might influence the representativeness of the cohort towards the general population. Indeed, the participants are relatively highly educated and have a relatively healthy lifestyle, as was shown by the low number of smokers and the in general low alcohol consumption [14, 15]. Additionally, compared to the invited or flagged, non-participating patients, participants in the Nivel Corona Cohort were slightly older, more often female and had less frequently lung diseases.

## Conclusion

The Nivel Corona Cohort provides ample opportunities for future studies to gain a comprehensive insight in the short- and long-term effects of COVID-19 in a population-based cohort by combining patients' perspectives and clinical parameters within a long-term follow-up of the cohort.

The present study shows that fatigue, reduced condition and coughing, sneezing and/or stuffy nose are frequent symptoms at the acute phase of COVID-19 in the general population with heterogeneous severity of symptoms.

## Supporting information

**S1 Fig. Prevalence of symptom burden marked to severe.**
(DOCX)

**S1 Table. Description of questionnaire and electronic health record data.**
(DOCX)

**S2 Table. Duration of the symptoms during the initial acute COVID-19 infection.**
(DOCX)

## Acknowledgments

The authors would like to thank all participants, patients as well as GPs, for their contribution to this study.

## Author Contributions

**Conceptualization:** Karin Hek, Laura Schackmann, Eugène van Puijenbroek, Liset van Dijk.

**Data curation:** Renee Veldkamp, Karin Hek.

**Formal analysis:** Renee Veldkamp.

**Funding acquisition:** Karin Hek, Liset van Dijk.

**Investigation:** Karin Hek.

**Methodology:** Karin Hek, Liset van Dijk.

**Project administration:** Renee Veldkamp, Karin Hek, Liset van Dijk.

**Supervision:** Karin Hek, Liset van Dijk.

**Visualization:** Renee Veldkamp.

**Writing – original draft:** Renee Veldkamp.

**Writing – review & editing:** Karin Hek, Rinske van den Hoek, Laura Schackmann, Eugène van Puijenbroek, Liset van Dijk.

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
