## [Decision Letter · Decision Letter 0]

4 Sep 2022

PONE-D-22-20156Combining general practice electronic records with patient reported outcomes to study impact of COVID-19 in patients in the general population: a description of the Nivel Corona CohortPLOS ONE

Dear Dr. Veldkamp,

Thank you for submitting your manuscript to PLOS ONE. After careful consideration, we feel that it has merit but does not fully meet PLOS ONE’s publication criteria as it currently stands. Therefore, we invite you to submit a revised version of the manuscript that addresses the points raised during the review process.

Your manuscript has been assessed by one peer-reviewer and their report is appended below.  The reviewer comments that the title of the manuscript does not appear to describe the nature of the study adequately. In addition, the reviewer highlights several areas of the study where the information or description provided is insufficient or unclear.  Please note that we have only been able to secure a single reviewer to assess your manuscript. We are issuing a decision on your manuscript at this point to prevent further delays in the evaluation of your manuscript. Please be aware that the editor who handles your revised manuscript might find it necessary to invite additional reviewers to assess this work once the revised manuscript is submitted. However, we will aim to proceed on the basis of this single review if possible. 

We look forward to receiving your revised manuscript.

Kind regards,

Maria Elisabeth Johanna Zalm, Ph.D

Editorial Office

PLOS ONE

Journal Requirements:

3. You indicated that you had ethical approval for your study. In your Methods section, please ensure you have also stated whether you obtained consent from parents or guardians of the minors included in the study or whether the research ethics committee or IRB specifically waived the need for their consent.

Reviewers' comments:

Reviewer's Responses to Questions

**Comments to the Author**

1. Is the manuscript technically sound, and do the data support the conclusions?

Reviewer #1: Partly

2. Has the statistical analysis been performed appropriately and rigorously? 

Reviewer #1: Yes

3. Have the authors made all data underlying the findings in their manuscript fully available?

Reviewer #1: Yes

4. Is the manuscript presented in an intelligible fashion and written in standard English?

Reviewer #1: Yes

5. Review Comments to the Author

Reviewer #1: The title of this manuscript does not coincide with the content of the manuscript, which is most on describing the results of the initial (recruitment) wave (‘Q1’) of the Nivel Corona Cohort (the patient reported outcomes). The theme of linking these outcomes with electronic records of patients is hardly addressed. As the authors state (page 4), the emphasis in this paper lays on describing the recruitment process, the representativeness and the characteristics of the cohort members, next to examining the (reported) symptoms and healthcare use of cohort members. It is thus suggested either to alter the title of the manuscript or to adapt its content.

Although on several occasions, the authors mention that the study “is representative of the general population” (in contrast with other approaches addressing only hospitalized patients), I have some serious questions on this or, at least, no strong arguments are presented to substantiate this claim.

First, the target population is not the general population, but the population of patients with a diagnosed infection. Second, the cohort recruitment lacks scientific soundness: it is stated that 25 (out of “approximally” 350) GP’s “that had already shown their interest in the study” (based on an early enquiry in May 2020) were contacted, of which 17 participated. It is not clear what “this early enquiry” was about. I do not understand the rationale to restrict the study to a these low numbers of GP’s – given that data-collection among patients is online (without substantial additional costs with increasing number of patients invited for participation to the study). The fact that the practices of the GP’s were spread throughout the Netherlands and were located in both rural and urban areas, looks to be a coincidence and look to be not very relevant in the context of the study. Third, as shown in table 1, the characteristics of the recruited cohort members in terms of age-structure and gender-structure are significantly different compared to the invited non-participants [the comparison with the total group of selected patients does not make much sense, since the difference between ‘selected’ and ‘invited’ is related to a rather arbitrary judgement of the GP on the eligibility of patients to be invited for participation].Given the unbalance between the composition of the cohort members and the composition of the invited patients, why was not a kind of (re-)weighting procedure applied?

As patients were recruited for participating to the cohort by means of an invitation letter and an access to an online application, patients not having access to an online application or not having the skills to access the application are – by default – excluded from participation. It looks like no specific efforts have been made to reach these patients by e.g. offering an alternative such as a paper questionnaire. It looks no efforts where made to enhance participation by e.g. sending a reminder. Can the authors comment on this?

Except from RES, four questions from SS12-I and SF-12, it looks like most questions used in Q1 were home-made questions. Given the ‘combining records and patient outcomes’ emphasis, I wonder why e.g. a question on” with how much certainty patients had COVID-19 and how long ago this was” is weird: (a) all patients were diagnosed as being infected by COVID-19 (ICPC-code R83.03) since this was a selection criterion, (b) all patients were evaluated for eligibility for participation by the GP’s. The time of onset/diagnosis can be retrieved from the electronic records (I guess). So what is the use of this question. The same, partially ,goes for the question on the severity of the complaints: the response category ‘needed to be hospitalized’ can (I guess) be retrieve from the electronic records. Also the time between the infection and the response on Q1 can be calculated partially based on data derived from the electronic records. Can the authors comment on this?

In ‘strengths and weaknesses’ part of the paper it is stated that ‘participants are relatively high educated and have a relatively healthy lifestyle, as was shown by the low number of smokers and the in general low alcohol consumption’ which (a) contradicts the presumed representativity of the cohort and (b) it is not known what basis was used to state that among cohort members the prevalence of smoking and alcohol is low (what is used as reference). Can the authors comment on this?

In general terms, this manuscript would benefit from a thorough revision, using clear-cut definitions on acute and long COVID, a scientific more sound description of the recruitment of the cohort, should provide more evidence for the presumed representativity and should more emphasize the usefulness and added value of linking electronic patient records with patient reported outcomes.

Language check is needed.

Detail:

P3: Please provide a clear-cut definition of long COVID with reference (WHO, NICE,…).

P4: Which (and why) clinical parameters derived from the EHRs will be used?

P5: As this paper is framed in the context of the Netherlands, it is useful to provide some figures on the number of diagnosed infections/deaths due to COVID 19 in the Netherlands.

P7: The cohort recruitment lacks scientific soundness: it is stated that 25 (out of “approximally” 350) GP’s “that had already shown their interest in the study” (based on an early enquiry in May 2020) were contacted, of which 17 participated. It is not clear what “this early enquiry” was about. I do not understand the rationale to restrict the study to a these low numbers of GP’s – given that data-collection among patients is online (without substantial additional costs with increasing number of patients invited for participation to the study). The fact that the practices of the GP’s were spread throughout the Netherlands and were located in both rural and urban areas, looks to be a coincidence and is not very relevant in the context of the study.

P7: In Figure 1 it is mentioned that COVID tests were performed by the GP’s OR by a testing facility (the testing facility informed the GP’s). Reference to this should be mentioned in the text.

P8: The ‘eligibility check’ of the flagged patients by the associated GP’s is scientifically not very sound and I do not understand the usefulness of it. GP’s could indicate that patients did not had COVID-19, while flagging the patients is based on IPC-code R83.03? Can the authors comment on this? Again, for data-collection an online approach is used, without substantial costs with increasing number of invitees.

P8: It is mentioned that ‘If a patient decided to participate, they could anonymously sign up online using a personal registration number that was provided in the letter.’: ‘anonymously’ contradicts with ‘using a personal registration number’….

P10 The phrase ‘Therefore, Q1 does not refer to the beginning of the COVID-19…’ is unclear. What does ‘the beginning of the COVID-19’ mean?

P10 How was dealt with item missingness voor RES and SSL12-I is too much detail in the context of this paper. Wat is meant by listwise deletion in case multiple answers were missing?

P11 What is known is that the patients were diagnosed with COVID-19, either by the GP or the testing centre. This is the basis of the selection. Where does the question on the initial severity of their complaints stem from? ‘Needed to be hospitalized’ = was hospitalized? Is this info not available in the EHR?

P11 Participants were asked to note for a list of symptoms with which severity they experienced this symptom during the acute phase of the infection (none, mild, moderate, marked, severe or I do not know).

P11 Is the ‘acute phase of the infection’ defined somewhere? Can the authors comment on this? What does the ‘past four weeks’ mean? For patients with recent COVID-19 diagnosis close to moment of completing the Q1 questionnaire, this might by difficult to answer.

P12 Imagine a patient was diagnosed with COVID very early in the Q1 reference period (6 months). ‘The month before ‘they got COVID-19’: 7 months before completing the Q1 questionnaire. The past month = 1 month before completing the Q1 questionnaire. Imagine a patient was diagnosed with COVID very late in the Q1 reference period (6 months). ‘The month before ‘they got COVID-19’: 1 month before completing the Q1 questionnaire, but the past month = 1 month before completing the Q1 questionnaire. Are these data comparable?

P13 The main success factor for this article is linking EHR-data with patient reported data. What is derived from EHR-data should be detailed (not ‘and number of chronic diseases’). What is the rationale behind selecting the listing ICPC – codes? Guess the EHR provides much more info, also related to COVID-19 diagnosis? For the moment the ‘contribution’ of EHR-data is quite poor.

P14 If I am correct, the selected patients are those patients that were flagged, the invited patients are those patients that where not set as non-eligble by the GP (and thus patients that received an invitation), while the nivel cohort are those patients that participated.

I do not think it makes much sense to report on the selected patients, since this is a very heterogeneous category including also patients without COVID diagnosis (according to the GP), deceased people,…

P15 I cannot find the figure of 1,851 in table 1!

P16 The figures in table 2 are not the same as in table 1, e.g. average age is 51.4 (± 13.8) in table 1! Do you mean that for 442-439 = 3 patients, the age is missing?

P19 So, only a minority (9.5%) contacted a GP after the first symptoms of the infection, while the majority of 76% contacted a GGD. Guess they were tested at the GGD and the info was uploaded in the EHR? Guess there is a difference between contacting a GGD or a GP (for testing) and seeking information on the RIVM website.

P20 Well, of the 441 respondent, 20.4% indicated to have had none or hardly any complaints. Guess these patients did not need aftercare…

6. PLOS authors have the option to publish the peer review history of their article (what does this mean?). If published, this will include your full peer review and any attached files.

Reviewer #1: No

---

## [Author Response · Author response to Decision Letter 0]

19 Oct 2022

Response letter - Revision PlosONE

Combining general practice electronic records with patient reported outcomes to study impact of COVID-19 in patients in the general population: a description of the Nivel Corona Cohort

General

We would like to thank the reviewer and editor for the comments that helped improving the manuscript and the opportunity to resubmit the manuscript. Please find our answers to the remarks below.

Reviewer #1

Comments

1. The title of this manuscript does not coincide with the content of the manuscript, which is most on describing the results of the initial (recruitment) wave (‘Q1’) of the Nivel Corona Cohort (the patient reported outcomes). The theme of linking these outcomes with electronic records of patients is hardly addressed. As the authors state (page 4), the emphasis in this paper lays on describing the recruitment process, the representativeness and the characteristics of the cohort members, next to examining the (reported) symptoms and healthcare use of cohort members. It is thus suggested either to alter the title of the manuscript or to adapt its content.

Response 1.: we understand the remark of the reviewer as indeed this manuscript does not describe results of the linking between the electronic health records (EHRs) and patient reported outcomes. Considering the complexity of the methodology used and the vast amount of data that can be described, the aim of this paper was mainly to describe the process and infrastructure used to recruit participants and to give a first description of this cohort. Therefore we had added “a description of the Nivel Corona Cohort” to the title. As the possibility to link the data with EHRs is one of the merits of this study design, we think it is important to mention this in the title. However, based on the remarks of the reviewer the title has been altered to:

“Nivel Corona Cohort: a description of the cohort and methodology used for combining general practice electronic health records with patient reported outcomes to study impact of COVID-19”

Furthermore, the aims of the paper have been rephrased in order to make these clearer for the readers:

“The aim of this study was twofold. The first aim was to describe the methodology and infrastructure used in the study to recruit individuals with COVID-19 and the representativeness of the population-based cohort. The second aim was to describe the characteristics of the population-based cohort of COVID-19 patients, their symptoms, and healthcare usage for COVID-19 and to examine whether demographical and clinical characteristics differed between patients who perceived the severity of their infection differently.”

2. Although on several occasions, the authors mention that the study “is representative of the general population” (in contrast with other approaches addressing only hospitalized patients), I have some serious questions on this or, at least, no strong arguments are presented to substantiate this claim.

2.1 First, the target population is not the general population, but the population of patients with a diagnosed infection.

2.2 Second, the cohort recruitment lacks scientific soundness: it is stated that 25 (out of “approximally” 350) GP’s “that had already shown their interest in the study” (based on an early enquiry in May 2020) were contacted, of which 17 participated. It is not clear what “this early enquiry” was about. I do not understand the rationale to restrict the study to a these low numbers of GP’s – given that data-collection among patients is online (without substantial additional costs with increasing number of patients invited for participation to the study). The fact that the practices of the GP’s were spread throughout the Netherlands and were located in both rural and urban areas, looks to be a coincidence and look to be not very relevant in the context of the study. 

2.3 Third, as shown in table 1, the characteristics of the recruited cohort members in terms of age-structure and gender-structure are significantly different compared to the invited non-participants [the comparison with the total group of selected patients does not make much sense, since the difference between ‘selected’ and ‘invited’ is related to a rather arbitrary judgement of the GP on the eligibility of patients to be invited for participation].Given the unbalance between the composition of the cohort members and the composition of the invited patients, why was not a kind of (re-)weighting procedure applied?

Response 2.: we thank the reviewer for this thorough comment. With regard to the first point, we meant to state that our cohort is drawn from the general population, as the Nivel Primary Care Database contains EHRs of all patients enlisted in the participating general practices and in the Netherlands all citizens are enlisted in one general practice. Therefore, no first selection was made on the population from which we recruited participants. However, we do understand the reviewers remark and therefore rephrased this point on multiple places in the manuscript from “the general population” to “population-based”, for example in the Discussion-section:

“The Nivel Corona Cohort consists of a heterogeneous group of persons that had COVID-19. It is a population-based cohort, as the participants were recruited via EHRs of GPs and all Dutch inhabitants are registered in one general practice. The Nivel Corona Cohort thereby represents individuals with COVID-19 in the general population by containing a complete range of number and severity of symptoms. The cohort provides the opportunity to study the short- and long-term effects of an infection with COVID-19 in a population-based sample instead of mainly in patients who were hospitalized.”

Considering the second point, although data collection of patients is indeed online, the initial invitation was sent on paper and patient selection requires some time and effort from general practitioners (GPs) and more so from Nivel staff. Furthermore, all data between Nivel and the GP for patient recruitment have to be transferred through a TTP to ensure privacy. Therefore, the assumption that more patients does not cost more money is not correct and the choice of including 18 GPs was limited by time and budget. The prior enquiry comprised a short questionnaire to assess whether practices were interested in participating and thus whether the study was feasible. A limited number of practices that replied positively to this enquiry were then invited, as we calculated that this would be enough to end up with a sample of 250 patients, the original number of patients we aimed at. As the number of COVID-19 patients quickly increased during the time of recruitment, we ended the recruitment phase with considerably more than 250 patients. The 25 practices that were initially invited had complete morbidity data in 2019 and weekly data over 2020, used ICPC-code R83.03 for patients with COVID-19, and had sufficient COVID-19 cases. Indeed location of the practices in rural and urban regions is more a logical consequence as we selected from a pool of practices throughout the country, than a selection criterion on its own. We do think however that it is relevant to know where practices (and thus patients) resided. We have now elaborated this explanation in the Method-section as follows: 

“In an early enquiry in May 2020 among practices participating in Nivel-PCD to assess feasibility of the study, 90 practices had already shown their interest in the study. A selection of 25 general practitioners (GPs) were invited via e-mail to participate in this study in February 2021. Practices were selected based on completeness of morbidity data in 2019, having weekly data in 2020, using R83.03 to code COVID-19 cases and having sufficient COVID-19 cases.”

Regarding the third point, we have removed the comparisons and information regarding the group of “selected” patients based on the reviewer’s remark. Furthermore, although there are significant differences in average age and gender distribution between the groups of participating patients and invited, but not participating patients, these differences are rather small. Therefore, we do not believe these to greatly affect the findings in the study. Furthermore, as we only report on the group that participates in the cohort we do not think a (re-)weighting procedure is needed. However, we have added the noted points of a significant, albeit small, difference in age and gender between these groups in the Discussion-section:

“[…]. This might influence the representativeness of the cohort towards the general population. Indeed, the participants are relatively highly educated and have a relatively healthy lifestyle, as was shown by the low number of smokers and the in general low alcohol consumption. Additionally, compared to the invited, non-participating patients, participants in the Nivel Corona Cohort were slightly older, more often female and had less frequently lung diseases.”

3. As patients were recruited for participating to the cohort by means of an invitation letter and an access to an online application, patients not having access to an online application or not having the skills to access the application are – by default – excluded from participation. It looks like no specific efforts have been made to reach these patients by e.g. offering an alternative such as a paper questionnaire. It looks no efforts were made to enhance participation by e.g. sending a reminder. Can the authors comment on this?

Response 3.: the reviewer makes a valid remark. A reminder was sent to patients three weeks after the invitation letter was sent, this is now added to the method section. Researchers themselves do not have the possibility to contact specific patients as the data set-up available to Nivel is designed to make it impossible to trace back to an individual. In the BRIMM-infrastructure participating GPs have via the thrusted third party the opportunity to send invitations to patients. However, the option to use paper questionnaires was indeed not offered, mainly because of budgetary reasons. This could have increased the response rate, particularly in specific groups. We have added these points in the limitations section of the Discussion:

“Lastly, as potential participants were judged for eligibility by their GP, the recruitment depended on the personal views of the GP. Additionally, we did not offer the alternative of paper questionnaires. Therefore, some groups were not represented by the cohort (e.g. those illiterate or digital illiterate, not sufficient proficient in the Dutch language, too severe disease burden). This might influence the representativeness of the cohort towards the general population.”

And in the methods section: 

“The TTP sent an invitation letter on paper on behalf of the GP, containing information on the study and the question to participate. A reminder letter on paper was sent three weeks after the invitation.”

4. Except from RES, four questions from SS12-I and SF-12, it looks like most questions used in Q1 were home-made questions. Given the ‘combining records and patient outcomes’ emphasis, I wonder why e.g. a question on” with how much certainty patients had COVID-19 and how long ago this was” is weird: (a) all patients were diagnosed as being infected by COVID-19 (ICPC-code R83.03) since this was a selection criterion, (b) all patients were evaluated for eligibility for participation by the GP’s. The time of onset/diagnosis can be retrieved from the electronic records (I guess). So what is the use of this question. The same, partially, goes for the question on the severity of the complaints: the response category ‘needed to be hospitalized’ can (I guess) be retrieve from the electronic records. Also the time between the infection and the response on Q1 can be calculated partially based on data derived from the electronic records. Can the authors comment on this?

Response 4.: We understand that these added questions seem redundant with the availability of the EHRs. However, the emphasis of the current paper was on the description of the used infrastructure and the Nivel Corona Cohort, and not yet on the data from the EHRs (see also response 1). Therefore, almost all results in the current paper stem from the first questionnaire and are patient-reported outcomes. More importantly, not all information is available in the EHRs. To elaborate on the specific questions of the reviewer: the ICPC-code used to flag patients with COVID-19 in the database (R83.03) was relatively new at the time of the study. Therefore, there was not much knowledge on how reliable the registration of the code was. Furthermore, the codes are not registered in function of the study, but are routinely registered data. Thus, a wrong code could for example have been registered by the GP. Additionally, while the EHRs from general practices contain a lot of information on the morbidity and health history of patients, they do not contain hospital data and therefore it is not possible to retrieve from the data whether someone was hospitalized. Therefore, we first asked the participants themselves in the first questionnaire (Q1) whether or not they had had COVID-19 and additionally how much time ago and with which severity. Additionally, the latter was also asked because we are interested in the perception of patients (e.g., the perceived severity) and how this impact outcomes such as quality of life. 

5. In ‘strengths and weaknesses’ part of the paper it is stated that ‘participants are relatively high educated and have a relatively healthy lifestyle, as was shown by the low number of smokers and the in general low alcohol consumption’ which (a) contradicts the presumed representativity of the cohort and (b) it is not known what basis was used to state that among cohort members the prevalence of smoking and alcohol is low (what is used as reference). Can the authors comment on this?

Response 5.: based on this and other comments of the reviewer we have downsized the emphasis on representativity by removing ‘in the general population’ from the title, by changing “representative of the general population” towards “population-based cohort” and by expanding on these points in the Discussion-section. Furthermore, we have added the reference of the Trimbos-institute for the prevalence of smoking and alcohol consumption in the Netherlands. The Trimbos-institute is a scientific institute with focuses on mental health, alcohol consumption, smoking and drug use in the Netherlands. In 2021, 20.6% of the Dutch inhabitants of 18 years and older smoked, in our cohort this was only 5.0%1. The difference for alcohol consumption is smaller. The percentage of individuals drinking no alcohol to one glass of alcohol per day was 48.6% in our cohort and 44.4% in the Dutch adult population. A smaller percentage of participants was however an excessive or heavy drinker, respectively 3.8% versus 15.6%2.

6. In general terms, this manuscript would benefit from a thorough revision, using a) clear-cut definitions on acute and long COVID, b) a scientific more sound description of the recruitment of the cohort, c) should provide more evidence for the presumed representativity and d) should more emphasize the usefulness and added value of linking electronic patient records with patient reported outcomes.

Response 6.: we have revised the manuscript based on the reviewers comments:

a. We have added the following definition for long COVID to the Introduction-section:

Post COVID syndrome (WHO): “PCS is defined by the WHO as the presence of COVID-19-related symptoms three months after the onset of COVID-19, which have occurred at least two out of those three months and cannot be explained by an alternative diagnosis 3.”

With the ‘acute phase of the infection’ the initial period after the infection was meant. In the questionnaire this was referred to as ‘at the time of infection’. Therefore, we have added the following definition for the acute phase of the COVID-19 infection to the Method-section:

Acute COVID: “the initial period following the time of infection.”

b. We have rewritten the description of the recruitment of the cohort, as described under comment 2.

c. We have reworked the manuscript with regard to the representativity, as described under comments 2 and 5, by changing “general population” to “population-based” and by elaborating on the reported differences affecting representativity.

d. The Nivel Primary Care Database contains routinely registered data from general practices and is a longitudinal database providing medical history of the patients. The type of data available includes for example diagnoses, drug prescriptions, lab measurements and information on primary health care usage as consults with the general practitioner. However, as the database contains data from general practices, no information on health care in the hospital is available. We have elaborated on the possibilities of the infrastructure used in the present study that allows linking of the EHRs with the patient reported outcomes and on the upcoming future studies that use the Nivel Corona Cohort in the Discussion section:

“Therefore, future papers using this cohort will focus on the course of the symptoms and their effects on quality of life, employment, and clinical information from EHR on prescriptions, GP consultations, referrals and lab measurements. Furthermore, using EHR data allows for example for analysis of the impact of prior medical history on the severity of COVID-19 and on the relation with quality of life. In addition, care pathways can be traced and longitudinal patterns of symptoms presented in general practice can be described.”

7. Language check is needed.

Response 7.: a native speaker performed a language check and changes have been made accordingly.

Detail

8. P3: Please provide a clear-cut definition of long COVID with reference (WHO, NICE,…).

Response 8.: we added the definition of post-acute COVID-19 syndrome to the Introduction-section:

Post COVID syndrome (WHO): “PCS is defined by the WHO as the presence of COVID-19-related symptoms three months after the onset of COVID-19, which have occurred at least two out of those three months and cannot be explained by an alternative diagnosis.”

9. P4: Which (and why) clinical parameters derived from the EHRs will be used?

Response 9.: Currently, our research group studies the prior medical history of patients in relation to the perceived burden of COVID-19 and to the impact COVID-19 has on quality of life. With this we want to study whether persons who presented more diseases and complaints in general practice are impacted differently by COVID-19 than those who have less diseases and symptoms. In due time, we will also study longitudinal disease and symptom patterns (based on EHR) among those who had more severe COVID-19 and who had PCS, compared to those with mild complaints or without PCS. Additionally, we will study the biases that occur in using survey data versus data from EHRs in defining PCS. In the Discussion-section the text as mentioned in reply 6.d. is added.

10. P5: As this paper is framed in the context of the Netherlands, it is useful to provide some figures on the number of diagnosed infections/deaths due to COVID 19 in the Netherlands.

Response 10.: We have now provided some figures on the number of deaths due to COVID-19 and the number of diagnosed infections known in September 2021. We have chosen to report upon the numbers in September 2021, as in September all participants were recruited and rapid changes in numbers have occurred in the past year, which are less relevant for this specific study. We have added this information in the Discussion-section:

“At that moment in time, two million Dutch inhabitants had been diagnosed and registered with COVID-194 and approximately 32,000 people had died in the Netherlands with COVID-19 as registered or probable cause of death5.”

11. P7: The cohort recruitment lacks scientific soundness: it is stated that 25 (out of “approximally” 350) GP’s “that had already shown their interest in the study” (based on an early enquiry in May 2020) were contacted, of which 17 participated. It is not clear what “this early enquiry” was about. I do not understand the rationale to restrict the study to a these low numbers of GP’s – given that data-collection among patients is online (without substantial additional costs with increasing number of patients invited for participation to the study). The fact that the practices of the GP’s were spread throughout the Netherlands and were located in both rural and urban areas, looks to be a coincidence and is not very relevant in the context of the study.

Response 11.: we updated the method section to clarify this, see our reply to point 2.2.

12. P7: In Figure 1 it is mentioned that COVID tests were performed by the GP’s OR by a testing facility (the testing facility informed the GP’s). Reference to this should be mentioned in the text.

Response 12.: we have added the following to the Method-section:

“The data received by Nivel-PCD consists of routinely registered health care data by the GP, among which diagnoses. International Classification of Primary Care (ICPC) codes are used to code diagnoses. The ICPC-code R83.03 was introduced by the Dutch College of General Practitioners to register COVID-19 from November 2020 onwards. The diagnosis of COVID-19 for an individual patient could be in their EHR when the patient consulted their GP directly, or when the patient contacted the Municipal Health services (GGD), who provided the national testing facilities. The GGD sent information on positive tests to GPs via automated coupling under the prerequisite that patients gave consent.”

13. P8: The ‘eligibility check’ of the flagged patients by the associated GP’s is scientifically not very sound and I do not understand the usefulness of it. GP’s could indicate that patients did not had COVID-19, while flagging the patients is based on IPC-code R83.03? Can the authors comment on this? Again, for data-collection an online approach is used, without substantial costs with increasing number of invitees.

Response 13.: Although the ‘eligibility check’ by the GP introduces some selection, it is an important step in the process that was introduced because of ethical considerations. First of all, the EHR is not made for research purposes and may include some registration errors. This check gives GPs the opportunity to check the correctness of the data. Additionally, not all relevant information is included in codes and the check gives GPs the opportunity to exclude patients of whom the GP knows that recently changes have occurred, such as death or moving (in which case the patient also transfers to another GP). Also, invitation letters are sent on behalf of the GP, so the GP should have the opportunity to check the selection. This is thus an important step to maintain credibility of the GP and a relation based upon trust between the GPs and our institute. Last, we did provide a standardized list of reasons for exclusion to the participating GPs, although the GPs indeed also had the possibility to provide other reasons than the ones listed. The following is mentioned in the Method-section:

“Patients were excluded when they were not proficient in the Dutch language, had passed away, were terminally ill, had moved or when they were judged by their GP as not having had COVID-19 or not being capable of filling out questionnaires, for example due to cognitive deficits, illiteracy, personal problems or too severe disease burden. The GPs were asked to register the reason for exclusion.”

14. P8: It is mentioned that ‘If a patient decided to participate, they could anonymously sign up online using a personal registration number that was provided in the letter.’: ‘anonymously’ contradicts with ‘using a personal registration number’….

Response 14.: this is a valid comment of the reviewer. We had written anonymously as there is no way for the researchers to trace back the individual patient. However, pseudonymized would be more correct and we have changed this sentence accordingly.

“Patients could participate using a personal registration number provided by the TTP which was not available to the researchers, who only received pseudonymized data.”

15. P10 The phrase ‘Therefore, Q1 does not refer to the beginning of the COVID-19…’ is unclear. What does ‘the beginning of the COVID-19’ mean?

Response 15.: by ‘the beginning of the COVID-19’ the onset of the disease in the patient was meant, i.e. the time of infection. We have rephrased the sentence:

“Therefore, Q1 does not refer to the onset of the COVID-19 infection for the patient, but to the beginning of participation of the patient (i.e., between zero and six months after infection).”

16. P10 How was dealt with item missingness for RES and SSL12-I is too much detail in the context of this paper. Wat is meant by listwise deletion in case multiple answers were missing?

Response 16.: we have deleted the last sentence to avoid too much detail. However, we left the sentence on imputation by the mean score in cases where only one answer on the questions of the RES or SSL12-I was missing, as this is a manipulation of the data that we think should be described.

17. P11 What is known is that the patients were diagnosed with COVID-19, either by the GP or the testing centre. This is the basis of the selection. Where does the question on the initial severity of their complaints stem from? ‘Needed to be hospitalized’ = was hospitalized? Is this info not available in the EHR?

Response 17.: See also our reply to comment 4. The participants were asked in Q1 to report on their perceived initial severity of symptoms and they could choose one of the following answers:

- “I had none to hardly any symptoms”

- “I had symptoms similar to a severe cold”

- “I had many symptoms, but did not have to go to the hospital”

- “I had so many symptoms that I was hospitalized”.

- “I don’t know / I don’t want to answer”

This is described in the Method-section of the manuscript, under the section “patient reported outcomes” and subsection “COVID-19, health care and symptoms”. This information on severity of symptoms is not available in the EHRs of the general practices and also concerns the perceived general severity of the symptoms by the patient. Also, we do not know from the EHR of the GP whether a patient was hospitalized. This type of information is normally conveyed to GPs via letters, where we only have coded data. Therefore, this is self-reported information by the patient. We did change ‘needed to be hospitalized’ into ‘was hospitalized’ and ‘complaints’ into ‘symptoms’ for clarity.

18. P11 Participants were asked to note for a list of symptoms with which severity they experienced this symptom during the acute phase of the infection (none, mild, moderate, marked, severe or I do not know). P11 Is the ‘acute phase of the infection’ defined somewhere? Can the authors comment on this? What does the ‘past four weeks’ mean? For patients with recent COVID-19 diagnosis close to moment of completing the Q1 questionnaire, this might by difficult to answer.

Response 18.: we have added a definition of the ‘acute phase of the infection’ as described under comment 6a. Furthermore, in the first questionnaire (Q1) some questions referred to the four-week period before the COVID-19 infection of the patients. These questions were added to obtain an image, although retrospectively, of for example the self-reported quality of life in the period before the patient got COVID-19. This was done in order to be able to examine the effect of the infection on that variable. Most questions however referred either to the experiences of the patient during the infection, so at the beginning of their infection, and to the most recent past four weeks. So with ‘past four weeks’ the four weeks prior to the participant filling the questionnaire and answering the questions were meant. We don’t think these questions regarding ‘the past four weeks’ are more or less difficult to answer for patients depending on the time since infection, as in all cases the past four weeks refer to the four weeks prior to filling the questionnaire. However, as researcher we do need to take the timeline into careful consideration while performing the analyses in the subsequent studies in which these type of data will be examined. This has been mentioned in the Discussion-section as follows:

“Furthermore, there was a broad range in the time between the infection and the first questionnaire, which needs to be accounted for when analyzing the post infection questions.”

19. P12 Imagine a patient was diagnosed with COVID very early in the Q1 reference period (6 months). ‘The month before ‘they got COVID-19’: 7 months before completing the Q1 questionnaire. The past month = 1 month before completing the Q1 questionnaire. Imagine a patient was diagnosed with COVID very late in the Q1 reference period (6 months). ‘The month before ‘they got COVID-19’: 1 month before completing the Q1 questionnaire, but the past month = 1 month before completing the Q1 questionnaire. Are these data comparable?

Response 19.: there was always some time between the COVID-19 infection of the patient and the completion of the Q1 questionnaire, as we are never allowed to approach patients from GPs directly because of reasons of privacy. Also, we are not allowed to know the registration number of the patients within the general practice, therefore pseudonymized numbers are used. As a result, the data first had to go through the full infrastructure (i.e., EHR data in Nivel-PCD � scanning by the algorithm � sending of flagged patients to GPs by the TTP � sending invitations letters to the eligible patients by the GP and � starting the questionnaire by the patient). Indeed, the median of time between the registered COVID-19 infection and Q1 was 28 days, with a minimum of 24 days. Furthermore, in the questionnaire the questions clearly referred to either the period before the COVID-19 infection of the patient, the symptoms at the time of infection and the four-week period prior to completing the questionnaire. Therefore, the situation that the questions on 1 month before the COVID-19 infection refer to exactly the same period as the past month, as stated in the second-last sentence of the comment, did not occur.

However, we do agree with the reviewer that the 1 month before the COVID-19 infection is a bit more problematic, as for some of the participants this period was quite a long time ago, which induces recall bias. We did however check whether the answers on these questions differed between groups that differed in the period between the time of infection and completing Q1, and this was not the case. Furthermore, we will take the differences in timelines into account in further analyses. The following is mentioned in the Discussion-section:

“However, some limitations of the current cohort should be taken into account. […] Secondly, for some questions respondents were also asked to describe the situation before the infection, as for example the SF12. Therefore, some recall bias might be present. Furthermore, there was a broad range in the time between the infection and the first questionnaire, which needs to be accounted for when analyzing the post infection questions. However, the answers concerning the time before infection (pre) did not differ between groups based on the time since infection.”

20. P13 The main success factor for this article is linking EHR-data with patient reported data. What is derived from EHR-data should be detailed (not ‘and number of chronic diseases’). What is the rationale behind selecting the listing ICPC – codes? Guess the EHR provides much more info, also related to COVID-19 diagnosis? For the moment the ‘contribution’ of EHR-data is quite poor.

Response 20.: we do agree with the reviewer that we should elaborate in the cases the EHR-data was used. Therefore, we added the reference for our selection of chronic diseases used for the construction of the variable ‘number of chronic diseases’ and the rationale behind the listed ICPCs:

“Age, gender, and number of chronic diseases as defined by Nielen et al (2016)6 were drawn from the EHRs. Additionally, the presence of comorbidities that were known risk factors for more severe COVID, was based on registered ICPC-codes in the EHRs, namely diabetes mellitus (ICPC T90), fat metabolism disorder (ICPC T93), lung disease (ICPCs R28, R91, R95, R96), and hypertension (ICPC K86, K87).”

Indeed in this particular manuscript the contribution of the EHR-data is poor. However, this manuscript was meant as an explanation of the methodology and infrastructure used to be able to link the patient reported outcomes and the EHRs, and additionally give a first, general, description of the cohort by describing findings from the first questionnaire (Q1) (see also our reply to comment 1). Subsequent studies will focus on the more in-depth possibilities of the current study-design, as the linking with data from EHRs and the patterns over time using also data from the other questionnaire (Q1, Q2, Q3 and Q4) depending on the topic (see also response 6d and 9). In these subsequent manuscripts the current paper is planned to be used as a basis. We have elaborated on this, by rephrasing the aims of the current paper in the Introduction-section and by adding some further explanations in the Discussion-section:

Introduction: “The aim of this study was twofold. The first aim was to describe the methodology and infrastructure used in the study to recruit individuals with COVID-19 and the representativeness of the population-based cohort. The second aim was to describe the characteristics of the population-based cohort of COVID-19 patients, their symptoms, and healthcare usage for COVID-19 and to examine whether demographical and clinical characteristics differed between patients who perceived the severity of their infection differently.”

Discussion: “Therefore, future papers using this cohort will focus on the course of the symptoms and their effects on quality of life, employment, and clinical information from EHR on prescriptions, GP consultations, referrals and lab measurements. Furthermore, using EHR data allows for example for analysis of the impact of prior medical history on the severity of COVID-19 and on the relation with quality of life. In addition, care pathways can be traced and longitudinal patterns of symptoms presented in general practice can be described.”

21. P14 If I am correct, the selected patients are those patients that were flagged, the invited patients are those patients that were not set as non-eligble by the GP (and thus patients that received an invitation), while the nivel cohort are those patients that participated.

I do not think it makes much sense to report on the selected patients, since this is a very heterogeneous category including also patients without COVID diagnosis (according to the GP), deceased people,…

Response 21.: the reviewer is correct in the interpretation of the groups. As described in response 13, we had good reasons to include the ‘eligibility-check’ by the GPs. Additionally, a standardized list of possible reasons for exclusion was provided to the GPs, although additional reasons based on the GPs own insights was possible. Our rationale for showing both the selected group of patients and the invited group of patients was to be as complete as possible. However, considering the reviewers valid remark on the heterogeneity of this group, including patients with according to the GP no COVID diagnosis, this group of ‘selected’/’flagged’ patients has now been removed from the comparisons in the results section. The participants are now only compared with the patients that were invited, but who did not participate. Changes in the Results and Method-sections have been made accordingly.

22. P15 I cannot find the figure of 1,851 in table 1!

Response 22.: that is correct, in total 1851 patients were invited of whom 442 participated. The comparisons made and shown in Table 1 are between the group of participants in the cohort and the group of patients that were invited, but did not participate: 1851 minus 442 = 1409 patients. This is mentioned in the footnote of the table (a). To make this clearer for the reader we have now also added this in text:

“Table 1 shows comparisons between the Nivel Corona Cohort (n=442) and the group of invited patients that did not participate (n= 1851–442 = 1409).”

23. P16 The figures in table 2 are not the same as in table 1, e.g. average age is 51.4 (± 13.8) in table 1! Do you mean that for 442-439 = 3 patients, the age is missing?

Response 23.: the reviewer is correct in this remark. The numbers in Table 1 are based on information drawn from the EHR as this age is compared to the age of the invited, but not participating patients. For this latter group we do not have questionnaire data as this group did not participate. In the remainder of the manuscript we describe mainly the data drawn from the questionnaires: herein three persons did not want to give their year of birth. Therefore, there is a discrepancy between these numbers in Table 1 and Table 2.

However, we do see that this is confusing and as those three persons gave consent to couple with their EHR we decided to use the data on their year of birth from the EHRs to complement the questionnaire data. Both tables now provide the same numbers.

24. P19 So, only a minority (9.5%) contacted a GP after the first symptoms of the infection, while the majority of 76% contacted a GGD. Guess they were tested at the GGD and the info was uploaded in the EHR? Guess there is a difference between contacting a GGD or a GP (for testing) and seeking information on the RIVM website.

Response 24.: indeed, in the Netherlands individuals could contact their GP or the GGD in case of a positive self-test or when they suspected to have got COVID-19. As the GGD arranged the national testing facilities, most persons indeed went to these facilities, which is also shown in the data. The GGD could sent information on positive tests to GPs via automated coupling under the prerequisite that the individual gave consent. This has now been added to the text in the Method-section. Furthermore, the question in which participants could also answer with ‘seeking information on the RIVM website’ was specifically asking for the first action of the individual upon discovering or suspecting they had COVID-19. Therefore, the answer of ‘seeking information on the RIVM website’ does not mean they did not contact their GP or the GGD for testing thereafter. 

“The data received by Nivel-PCD consists of routinely registered health care data by the GP, among which diagnoses. International Classification of Primary Care (ICPC) codes are used to code diagnoses. The ICPC-code R83.03 was introduced by the Dutch College of General Practitioners to register COVID-19 from November 2020 onwards. The diagnosis of COVID-19 for an individual patient could be in their EHR when the patient consulted their GP directly, or when the patient contacted the Municipal Health services (GGD), who provided the national testing facilities. The GGD sent information on positive tests to GPs via automated coupling under the prerequisite that patients gave consent.”

25. P20 Well, of the 441 respondents, 20.4% indicated to have had none or hardly any complaints. Guess these patients did not need aftercare…

Response 25.: that is one of the research questions that will be answered. Also patients with no or hardly any complaints may develop PCS. Therefore, subsequent studies will perform more in-depth examinations of the data on this topic for which the EHR data will be used. Our aim is to show what happened to all patients within this cohort, regardless of their initial perceived severity of complaints and need for aftercare.

Comments from the Journal editor:

The unmarked version of the revised manuscript (‘Manuscript’) has been updated to meet the PLOS ONE’s style requirements.

2. Please provide additional details regarding participant consent. In the ethics statement in the Methods and online submission information, please ensure that you have specified what type you obtained (for instance, written or verbal, and if verbal, how it was documented and witnessed). If your study included minors, state whether you obtained consent from parents or guardians. If the need for consent was waived by the ethics committee, please include this information. Once you have amended this/these statement(s) in the Methods section of the manuscript, please add the same text to the “Ethics Statement” field of the submission form (via “Edit Submission”). For additional information about PLOS ONE ethical requirements for human subjects research, please refer to http://journals.plos.org/plosone/s/submission-guidelines#loc-human-subjects-research.

The following has been added to the Method – section Medical ethical committee and changed accordingly in the “Ethics Statement” field of the submission form.

“All participants received written information on the study and gave digital informed consent. Participants could additionally provide informed consent for linkage of EHR data to questionnaire data. No consent from parents or guardians was needed for the minors in this study as they were all 16 years and older.”

3. You indicated that you had ethical approval for your study. In your Methods section, please ensure you have also stated whether you obtained consent from parents or guardians of the minors included in the study or whether the research ethics committee or IRB specifically waived the need for their consent.

In the Netherlands no consent of parents or guardians is needed from the age of 16 years on. The following has been added to the Method – section Medical ethical committee:

“No consent from parents or guardians was needed for the minors in this study as they were all 16 years and older.”

Access to data in the Nivel Primary Care Database is subject to Nivel Primary Care Database governance codes. Requests for access to the data can be directed at directie@nivel.nl (Nivel-PCD). Restrictions involve establishing a data sharing agreement and approval by the appropriate Nivel Primary Care Database governance bodies (privacy committee and steering committee). However, data concerning the used questionnaire in the present manuscript, which is the main body of data used, will be made accessible via DANS | Centre of expertise & repository for research data.

We have now included the captions of the three supporting information files at the end of the manuscript and changed the format to ‘S1 Figure. Title’, ‘S1 Table. Title’ and ‘S2 Table. Title’. In text citations have been adjusted to: “see S1 Fig”.

References

1. Bommelé J and Willemsen M. SMOKING IN THE NETHERLANDS: KEY STATISTICS FOR 2021, (2022).

2. Intitute T. Cijfers alcohol, https://www.trimbos.nl/kennis/cijfers/alcohol/ (2022).

3. WHO. A clinical case definition of post COVID-19 condition by a Delphi consensus, 6 October 2021, https://www.who.int/publications/i/item/WHO-2019-nCoV-Post_COVID-19_condition-Clinical_case_definition-2021.1 (2021).

4. RIVM. COVID-19 dataset - COVID-19_aantallen_gemeente_cumulatief, https://data.rivm.nl/covid-19/ (2022).

5. CBS. In 2021 ruim 19 duizend mensen aan COVID-19 overleden, https://www.cbs.nl/nl-nl/nieuws/2022/13/in-2021-ruim-19-duizend-mensen-aan-covid-19-overleden (2022, accessed 13-10-2022).

6. Nielen M, Davids R, Gommer M, et al. Berekening morbiditeitscijfers op basis van NIVEL Zorgregistraties eerste lijn. 2016. Utrecht: NIVEL.

---

## [Decision Letter · Decision Letter 1]

9 Feb 2023

PONE-D-22-20156R1Nivel Corona Cohort: a description of the cohort and methodology used for combining general practice electronic records with patient reported outcomes to study impact of COVID-19PLOS ONE

Dear Dr. Veldkamp,

Thank you for submitting your manuscript to PLOS ONE. After careful consideration, we feel that it has merit but does not fully meet PLOS ONE’s publication criteria as it currently stands. Therefore, we invite you to submit a revised version of the manuscript that addresses the points raised during the review process.

We look forward to receiving your revised manuscript.

Kind regards,

Stefaan Demarest

Guest Editor

PLOS ONE

Additional Editor Comments:

Compared to the previous (first) version of the manuscript, the authors have addressed most of the remarks. A supplementary review revealed that still some elements need to be clarified. Authors are asked to address the issues raised in this review.

Reviewers' comments:

Reviewer's Responses to Questions

**Comments to the Author**

1. If the authors have adequately addressed your comments raised in a previous round of review and you feel that this manuscript is now acceptable for publication, you may indicate that here to bypass the “Comments to the Author” section, enter your conflict of interest statement in the “Confidential to Editor” section, and submit your "Accept" recommendation.

Reviewer #2: (No Response)

2. Is the manuscript technically sound, and do the data support the conclusions?

Reviewer #2: No

3. Has the statistical analysis been performed appropriately and rigorously? 

Reviewer #2: No

4. Have the authors made all data underlying the findings in their manuscript fully available?

Reviewer #2: (No Response)

5. Is the manuscript presented in an intelligible fashion and written in standard English?

Reviewer #2: Yes

6. Review Comments to the Author

Reviewer #2: This study needs very extensive revision. The first aim of this manuscript is to describe the recruitment methodology, but the way it was presented was not satisfactory as the description was very qualitative while a clear step by step process with numbers attached would have been more transparent. The second aim was to describe the cohort and do some additional analysis. This part needs to be more focused as plenty of indicators were analyzed, but there is no clear rational for the inclusion of all this wide array of outcomes. Generally, in a cohort description, one can limit the focus to a limited number of carefully selected outcomes.

Specific comments

Title

Needs to be more specific as to the impact of COVID-19 on what ? health ? which dimension of health? Also, impact of the pandemic ? or a COVID-19 infection ?

Abstract

-Suggest to add subsections for better clarity.

Introduction

-In the Introduction, there is an important focus on post-acute COVID-19 syndrome (PCS). However, this issue is not examined further in the paper. So, this provides some confusion as to the aim of the paper.

-In the abstract, the second aim was : “to examine the symptoms and healthcare usage during the acute COVID-19 phase”. However, the second aim in the introduction is much more wide and does not mention the acute phase anymore : “The second aim was to describe the characteristics of the population-based cohort of COVID19 patients, their symptoms, and healthcare usage for COVID-19 and to examine whether demographical and clinical characteristics differed between patients who perceived the severity of their infection differently.” It is necessary therefore that the aim of the paper is clarified and is consistent between the different sections of the paper.

Method

-The recruitment process needs to be described in a more transparent manner. Figure 1 is clear but the authors need to attach the numbers to it. The process has started with how many people in order to arrive at the end of the process with 442 participants ?

-The description of the questionnaire is quite lengthy and messy, more info can go to a table and it would be clearer and more interesting to the readers to have a more condensed and synthetic information about the questionnaire and the electronic record, rather than so much details in the text.

-The authors need to add a rational for including and analyzing this series of variables.

Data analysis

-Representativeness is not only assessed compared to the population you have invited but to the target population, in this case for instance people who have the R83.03 code in the EHR.

-The subsection “analyses” was not clear to me.

Results

-In Q1, the cohort includes participants with acute COVID-19 (with or without symptoms), people with long COVID, and people who have recovered. When presenting the results for instance of quality of life, health care, social support, physical activity or resilience, do they refer to the current situation or the situation during the acute phase of the disease? If the acute phase, then how was it defined ? If the current situation, then the results would be mixing these groups with very different situations.

-Table 2 has a messy format, please rearrange- Please spell out SSL12 and RES.

-Again, as mentioned above, too many outcomes and no rationale for their inclusion. For instance, why social support, hours seated, alcohol consumption, etc.. a rational for including the different outcomes analyzed in the context of this manuscript is necessary.

-What is the rationale to have so detailed info on health care in Table 3?

-Severity: the authors starts the subsection with " During the acute COVID-19 phase". How is this defined ?

-Prevalence of symptoms: The proportions seems very high to me, so these are during the acute phase and only for people who declared symptoms ? Because in the Severity section, 20.4% of the people indicated to have had none or hardly any symptom. So, how do we have 90% with fatigue, 88% with reduced condition, etc…

-Aftercare: what is meant by aftercare ?

-Generally, there are too many subtitles, the subsections could be more integrated. For instance we do not need 3 subtitles in the data analysis section. Also, the information needs to be presented in a more integrated way as now it is very compartmentalized. For instance, why do we have a section on COVID-19 infection, severity and another about prevalence of symptoms and duration.

Discussion

-It was mainly a summary rather than a discussion

-No discussion or mention of first aim of the study.

7. PLOS authors have the option to publish the peer review history of their article (what does this mean?). If published, this will include your full peer review and any attached files.

Reviewer #2: No

---

## [Author Response · Author response to Decision Letter 1]

30 Jun 2023

General

We would like to thank the reviewer and editor for the comments that helped improving the manuscript and the opportunity to resubmit the manuscript. Please find our answers to the remarks below.

Additional Editor Comments

Compared to the previous (first) version of the manuscript, the authors have addressed most of the remarks. A supplementary review revealed that still some elements need to be clarified. Authors are asked to address the issues raised in this review.

Reviewer #2

General

This study needs very extensive revision. The first aim of this manuscript is to describe the recruitment methodology, but the way it was presented was not satisfactory as the description was very qualitative while a clear step by step process with numbers attached would have been more transparent. The second aim was to describe the cohort and do some additional analysis. This part needs to be more focused as plenty of indicators were analyzed, but there is no clear rational for the inclusion of all this wide array of outcomes. Generally, in a cohort description, one can limit the focus to a limited number of carefully selected outcomes.

Specific comments

1. Title: Needs to be more specific as to the impact of COVID-19 on what ? health ? which dimension of health? Also, impact of the pandemic ? or a COVID-19 infection ?

Response 1.: We updated our title to clarify that we study the impact of a COVID-19 infection. As we measure many different variables (work, health, quality of life) we do not further specify the title. 

2. Abstract: Suggest to add subsections for better clarity.

Response 2.: We added subsections for clarity reasons. 

3. In the Introduction, there is an important focus on post-acute COVID-19 syndrome (PCS). However, this issue is not examined further in the paper. So, this provides some confusion as to the aim of the paper.

Response 3.: The cohort was set up with the aim to investigate both the short- and long-term impact of a COVID-19 infection. In the introduction there was indeed a main focus on PCS. We agree with the reviewer that this is not in line with the aim of the paper and therefore shortened the paragraph on PCS.

4. Introduction: In the abstract, the second aim was : “to examine the symptoms and healthcare usage during the acute COVID-19 phase”. However, the second aim in the introduction is much more wide and does not mention the acute phase anymore : “The second aim was to describe the characteristics of the population-based cohort of COVID19 patients, their symptoms, and healthcare usage for COVID-19 and to examine whether demographical and clinical characteristics differed between patients who perceived the severity of their infection differently.” It is necessary therefore that the aim of the paper is clarified and is consistent between the different sections of the paper.

Response 4.: The aim of the paper is 1. to provide a description of the set-up of the cohort, including representativeness and 2. to describe the primary characteristics of the cohort on symptoms and health care usage in the acute phase. We updated the aim of the introduction to match the aim of the abstract. 

5. Method: The recruitment process needs to be described in a more transparent manner. Figure 1 is clear but the authors need to attach the numbers to it. The process has started with how many people in order to arrive at the end of the process with 442 participants ?

Response 5.: The first paragraph in the results section describes how we arrived at 442 participants. We chose to not include numbers in figure 1 to avoid adding results to the methods section. We restructured the paragraphs so that all information on participation is now included in one paragraph. 

6. Method: The description of the questionnaire is quite lengthy and messy, more info can go to a table and it would be clearer and more interesting to the readers to have a more condensed and synthetic information about the questionnaire and the electronic record, rather than so much details in the text.

Response 6.: We included two short paragraphs on patient reported outcomes and electronic health record data and included an additional supplementary table for the extensive description. 

7. Method: The authors need to add a rational for including and analyzing this series of variables.

Response 7.: We included these variables to be able to perform an indepth study of the short- and long-term effects of a COVID-19 infection. Therefore we included information on determinants (such as life style, but also resilience) and on outcomes, such as quality of life, and shortness of breath in questionnaires sent to the cohort

8. Data analysis: Representativeness is not only assessed compared to the population you have invited but to the target population, in this case for instance people who have the R83.03 code in the EHR.

Response 8.: The target population (flagged patients minus those that participate in the Nivel corona cohort) was now also compared to the population in the Nivel corona cohort in table 1. As only 252 of the 2103 flagged patients were excluded by the GPs, the characteristics of this population were very much like those of the invited population. 

9. Data analysis: The subsection “analyses” was not clear to me.

Response 9.: The data analysis paragraph was condensed into one paragraph including a description of both cohort representativeness and the cohort characteristics within the acute phase of the infection. 

10. Results: In Q1, the cohort includes participants with acute COVID-19 (with or without symptoms), people with long COVID, and people who have recovered. When presenting the results for instance of quality of life, health care, social support, physical activity or resilience, do they refer to the current situation or the situation during the acute phase of the disease? If the acute phase, then how was it defined ? If the current situation, then the results would be mixing these groups with very different situations.

Response 10.: For traits that are relatively stable we asked for the current status. For several other questions (such as health care use, health status, shortness of breath, quality of life) we made a distinction between before COVID-19 and at the moment of filling in the questionnaire. This is included in supplementary table 1A. In the results section we use data from Q1 (acute phase) and included in the headings and/or subtitles of tables where applicable. 

11. Results: Table 2 has a messy format, please rearrange- Please spell out SSL12 and RES.

Response 11.: Table 2 was rearranged and the abbreviations were spelled out. 

12. Results: Again, as mentioned above, too many outcomes and no rationale for their inclusion. For instance, why social support, hours seated, alcohol consumption, etc.. a rational for including the different outcomes analyzed in the context of this manuscript is necessary.

Response 12.: For this paper we aimed to characterize the cohort, we did not mean to be exhaustive. Therefore, in agreement with the comment of the reviewer, we made a selection of the presented results that are in line with our research questions. 

13. Results: What is the rationale to have so detailed info on health care in Table 3?

Response 13.: This was now removed from the paper to be concise and more in line with the research questions.

14. Results: Severity: the authors starts the subsection with " During the acute COVID-19 phase". How is this defined ?

Response 14.: This was based on self-report. In the questionnaire we asked for symptoms during the acute phase. This is now clarified in the methods section.

15. Results: Prevalence of symptoms: The proportions seems very high to me, so these are during the acute phase and only for people who declared symptoms ? Because in the Severity section, 20.4% of the people indicated to have had none or hardly any symptom. So, how do we have 90% with fatigue, 88% with reduced condition, etc…

Response 15.: Both questions were based on self-report. 90% reported to have had fatigue, however this consisted of mild, moderate, marked and severe fatigue. The latter was now included in the text. 

16. Results: Aftercare: what is meant by aftercare ?

Response 16.: We meant to describe care after the acute phase of the COVID-19 infection. However, this was now removed from the results section as it was not required to answer the research questions. 

17. Results: Generally, there are too many subtitles, the subsections could be more integrated. For instance we do not need 3 subtitles in the data analysis section. Also, the information needs to be presented in a more integrated way as now it is very compartmentalized. For instance, why do we have a section on COVID-19 infection, severity and another about prevalence of symptoms and duration.

Response 17.: The different sections on data analysis were now combined as were the different sections on the COVID-19 infection.

18. Discussion: It was mainly a summary rather than a discussion

Response 18.: The part where we summarized the results was condensed, to make it more a discussion, rather than a summary. 

19. Discussion: No discussion or mention of first aim of the study.

Response 19.: The first aim of the study was now included in the first paragraph of the discussion.

---

## [Editor Report · Decision Letter 2]

5 Jul 2023

Nivel Corona Cohort: a description of the cohort and methodology used for combining general practice electronic records with patient reported outcomes to study impact of a COVID-19 infection

PONE-D-22-20156R2

Dear Dr. Hek,

We’re pleased to inform you that your manuscript has been judged scientifically suitable for publication and will be formally accepted for publication once it meets all outstanding technical requirements.

Kind regards,

Stefaan Demarest

Guest Editor

PLOS ONE

Additional Editor Comments (optional):

The authors have addressed the issues raised by the reviewer and editor. This paper can be accepted for publication
---

## [Editor Report · Acceptance letter]

10 Aug 2023

PONE-D-22-20156R2 

Nivel Corona Cohort: a description of the cohort and methodology used for combining general practice electronic records with patient reported outcomes to study impact of a COVID-19 infection 

Dear Dr. Hek:

I'm pleased to inform you that your manuscript has been deemed suitable for publication in PLOS ONE. Congratulations! Your manuscript is now with our production department. 

Kind regards, 

on behalf of

Mr. Stefaan Demarest 

Guest Editor

PLOS ONE